# Phosphorus Availability and Balance with Long-Term Sewage Sludge and Nitrogen Fertilization in Chernozem Soil under Maize Monoculture

**DOI:** 10.3390/plants13152037

**Published:** 2024-07-24

**Authors:** Dinkayehu Alamnie Asrade, Martin Kulhánek, Jiří Balík, Jindřich Černý, Ondřej Sedlář, Pavel Suran

**Affiliations:** Department of Agroenvironmental Chemistry and Plant Nutrition, Faculty of Agrobiology, Food and Natural Resources, Czech University of Life Sciences, 165 00 Prague, Czech Republic; asrade@af.czu.cz (D.A.A.); balik@af.czu.cz (J.B.); cernyj@af.czu.cz (J.Č.); sedlar@af.czu.cz (O.S.); suranp@af.czu.cz (P.S.)

**Keywords:** maize, dry matter yield, sorption parameters, P availability, legacy P, P balance

## Abstract

A continuous long-term field experiment with maize monoculture was conducted to evaluate the P availability and balance, DM yield, P uptake, and P sorption parameters in chernozem soil after 27 years. A total of 2 doses of nitrogen (120 and 240 kg ha^−1^) were applied as mineral nitrogen (N_120_ and N_240_) and sewage sludge (SS_120_ and SS_240_) and compared with unfertilized control (Con). The aboveground biomass (DM) yields significantly increased in the order of Con < SS_120_ < SS_240_ < N_120_ < N_240_ treatments and the maximum P uptake was recorded for both N_240_ and SS_240_ (25.1 kg P ha^−1^) according to the nutrient application gradient. The N_120_ and N_240_ treatments positively influenced the DM yield but negatively influenced the P balance (−648 and −678 kg P ha^−1^ 27 years^−1^), gradually bringing a risk of P deficiency in the soil. On the other hand, applications of SS_120_ and SS_240_ positively influenced the P availability and pseudototal (P_AR_) content in the soil, which resulted in a buildup of legacy P or an increase in P saturation greater than the environmental threshold value. Aluminum was found to be a major controlling sorption factor for P in our chernozem soil.

## 1. Introduction

Phosphorus is one of the most acquired and limited plant nutrients in soil. About 29% of arable land is affected by insufficient P at a global level [1]. Phosphorus (P) in soil is mainly constituted of 30–50% of organic P and 35–70% of inorganic P [2,3]. However, only 10–30% of P is available for plant uptake and the remaining (70–90%) is intercepted by soil organic matter (SOM) (accounts for 20–50% of P sorption in both surface and subsurface soil), and metal (Fe, Al (accounted 37% of P sorption) and Ca) oxides depend on soil pH [3,4,5]. However, the intercepted P gradually became available through acquisition, mobilization, hydrolysis, mineralization, and decomposition processes in the root–soil–rhizosphere nexus. 

The nature of P cycling and transformation dynamics in soil is mainly dependent on varied physicochemical properties of soil (soil type and texture, bulk density, soil moisture, pH, SOM), agronomic practices (fertilizer type and rate, crop, and cropping systems), climate variability and agro-climatic regions (rainfall, temperature, precipitation, tropic, subtropic, and temperate), and locations (latitude and altitude) [6,7]. Other researchers also mentioned that the physico-chemical variability of soil, adsorption/sorption, and precipitation process are major controlling factors for both applied and legacy P, which leads to low P utilization (15–30%) in agriculture [8,9]. These are the predominant variables that determine soil nutrient cycling dynamics, especially for less mobile nutrients, i.e., phosphorus. Type of fertilizer like nitrogen-only fertilization also increased the mobilization of legacy P in soil [8]; therefore, it is vital to implement holistic soil P management and inventory approaches to estimate the entire P transformation from source to sink along its environmental impacts. Pre- and post-agroenvironmental soil nutrient status assessments are more important than ever to boost plant productivity and nutrient use efficiency, addressing environmental risks [10]. For example, Rowe et al. [8] have proposed five R (5R) key strategies/stewardship (realign P inputs, reduce P losses, recycle P from bioresources, recover P from wastes, and redefine crop P requirements) for sustainable P utilization. P inputs can be realigned based on crop yield, native soil nutrient status, and crop response to fertilizer [11]. When P is not a limiting nutrient in soil, N application is mostly recommended because of its dual role in making available the legacy P and meeting the N requirements of a plant. However, eventually, it brings a risk of P deficiency and depletion of legacy P [12], nutrient imbalance and plant growth, and yield limitation [13], which affect about 70% of croplands across the globe [14]. These limitations could be addressed through nutrient synchronization or by supplying single or combined N or NP. In fact, plants also try to correct the limitation by mobilizing other resources (i.e., carbon, enzymes, organic acids) and by adjusting their growth [14].

Generally, nutrient cycling in soil mainly depends on the capacity of a soil to (i) receive and retain nutrients, (ii) make or keep the nutrients, (iii) recover plant available nutrients, and (iv) support the amount of nutrients acquired by crop [7].

P transformation dynamics is widely reported in different studies with different agronomic practices, crop types, soil types, climatic regions, and other anthropogenic factors. In this study, a long-term field experiment with continuous silage maize monoculture was conducted to evaluate the P development status in Chernozem soil after long-term mineral (N) and organic (sewage sludge) fertilization because maize is one of the strong P scavengers of both soil P and applied P among cereal crops [15]. 

About 80% of the P fertilizer supply is from a limited non-renewable rock phosphate apart from other industrial uses [16]. This can be replaced with other alternative renewable sources, i.e., sewage sludge, which contains about 2.2–17.7% of P and 2.4–9% of N in dry matter [17]. Sewage sludge application also supplied 269 kg SOM ha^−1^year^−1^ [18], increased soil aggregate stability by 41% in sandy loam soil [19], increased pseudototal P by 31%, and increased available P_M3_ (67%) in Haplic Luvisol [20]. Sewage sludge has the potential to completely or partially replace mineral phosphorus. However, the degree and quantity of recovered P from SS depend on P recovery technologies, treatment types, and the potential of SS sources [16,21,22].

According to the global soil map, Chernozem covers 725 million ha (5.6%) of global land areas, of which 267, 227, and 212 million ha are covered by crops, forests, and grasses, respectively [23]. The dominant soil types in the arable land of the Czech Republic are Cambisols, Luvisols, Chernozem, Stagnosols, and Fluvisols. High-quality Chernozem soil is found in Southern Moravia and in the Northwestern Bohemia region [24]. As reported by Medinski et al. [4], Chernozem soils are characterized by low P sorption, high labile P, and naturally fertile soil compared to Ferralsol and Nitisol soils due to low formation of Al and Fe phosphate, high dissolution of Ca phosphates, and near-neutral soil pH (6–7) ranges. 

In our previous study, we tried to evaluate the silage maize biomass yield, P uptake, and nature of P transformation dynamics and soil P sorption capacity in Haplic Luvisol soil under long-term inorganic and organic fertilization with silage maize monoculture [12,20]. 

Thus, this study aimed to assess the P development change and N interactive effects on phosphorus mobility and transformation in Chernozem soil after long-term mineral and sewage sludge fertilization of silage maize monoculture.

## 2. Materials and Methods

### 2.1. Site Description

The long-term maize monoculture field experiment study has been conducted at Praha-Suchdol, 50°07′54″ N, 14°22′20″ E, in the Czech Republic since 1993 with the basic characteristics described in Table 1. The soil belongs to the silty loam group with high soil organic matter content as well as high pH value. The content of phosphorus measured at the beginning of the experiment was considered as high according to the evaluation criteria of the Central Institute of Supervising and Testing in Agriculture. The average yearly temperatures and the sum of yearly precipitations are highlighted in Figure 1. From the linear regression, it is obvious that, on one hand, the temperature was increasing, and on the other hand, the sum of precipitations was decreasing with time.

### 2.2. Experimental Design

The field experiment was carried out in a randomized block design with a 46 m^2^ plot size. The period evaluated in this manuscript was 27 years (1997–2023), comprising five treatments including the control, two doses of mineral nitrogen (N_120_, N_240_), and two doses of sewage sludge (SS_120_, and SS_240_) in four replications and with the following maize hybrids: TORENA (1997–1998), DK 254 (1999), COMPACT (2000–2001), ETENDARD (2002–2003), RIVALDO (2004–2011), RGT INDEXX (2012–2014), RGT SIXXTUS (2015–2020), and RGT ATTRAXXION (2021–2023), planted on each plot at a density of 80,000 plants ha^−1^. Site stabilization activity continued till 1996 without nutrient application. Fertilizer application began in the autumn of 1996 with the sewage sludge application. The mineral N (N_120_ and N_240_) fertilizers (calcium ammonium nitrate, 27% N) were applied prior to sowing every year (since 1997) in spring. SS_120_ and SS_240_ were applied once every three years in autumn according to the unified dose of N (120 kg ha^−1^ year^−1^ and 240 kg ha^−1^ year^−1^), which was set based on the Kjeldahl analysis of the fertilizers. The dose of total P applied in the form of sewage sludge was calculated based on the results of high-pressure microwave-assisted digestion. Sewage sludge was immediately incorporated into the soil by plowing to a depth of 25 cm. The pre-sowing soil preparation activities were started in spring, in the second week of March to the beginning of May, and were followed by sowing. Agronomic management practices such as weed, pest, and disease control were performed according to the actual situation in the field. The complete fertilizing design is presented in Table 2. Furthermore, sewage sludge contains a considerable amount of Al and Fe, which are used for phosphorus precipitation during wastewater processing. The yearly dose of Al was 46 kg ha^−1^ for SS_120_ and 92 kg ha^−1^ for SS_240_, respectively. The yearly dose of Fe was 117 kg ha^−1^ for SS_120_ and 234 kg ha^−1^ for SS_240_, respectively.

### 2.3. Determination of BY, P Uptake, and Total P Content in Aboveground Biomass

One experimental plot consisted of 4 rows of maize plants, which were 10 m long. The middle 2 rows (18 m^2^) were harvested at roughly 65% moisture (BBCH75 vegetation period) to obtain the BY. Representative subsamples (5 plants per plot) were mechanically chopped, dried at 40 °C for a minimum of 72 h, and fine-milled to <1 mm (Retsch SM100, Retsch GmbH, Haan, Germany) for subsequent analyses.

The total P content in the aboveground biomass was analyzed using the low-pressure microwave-assisted (HNO_3_-H_2_O_2_) digestion method of Mader et al. [29]. Briefly, about 0.4 g of dried aboveground biomass plant samples were digested with a mixture of 8 mL (HNO_3_ 68% *v*/*v*) and 2 mL of H_2_O_2_ (30% *v*/*v*) acid solution using a microwave-assisted wet digestion system under low pressure, and the amount of P concentration was analyzed using inductively coupled plasma optical emission spectroscopy (ICP-OES; Varian 720, Agilent technologies, Mulgrave, Australia). The phosphorus uptake was calculated from the dry biomass yield and P content in plants.

### 2.4. Soil Analysis

Archive soil samples from 1999, 2011, and 2023 were evaluated to see the P fraction content development.

The composite soil samples comprised 7 soil cores per plot collected from topsoil at 0–30 cm depth. Soil samples were then air-dried, ground, and sieved through a 2 mm sieve and analyzed for pH (CaCl_2_), available P (P_H2O_ (1/10 *w*/*v*)) [30], P, Fe, and Al, which were determined using Mehlich 3 extraction (P_M3_, Al_M3_, and Fe_M3_) (1/10 *w*/*v*) [28], and the pseudototal P (P_AR_) (1/2 *w*/*v*) was determined with *Aqua regia* extraction solution (3 HCl (37% *v*/*v*):1 HNO_3_ (65% *v*/*v*)) using a microwave-assisted wet digestion system (MLS GmbH, Leutkirch im Allgäu, Germany) under high pressure [27].

#### 2.4.1. Organic and Inorganic P Determination

Soil samples were ground to pass through a 0.1 mm size sieve for inorganic P determination according to the method in [31], where 0.6 g of the sample was shaken with 30 mL of 0.5 mol·L^−1^ H_2_SO_4_ for 16 h. This was followed by centrifugation to obtain the clean supernatant to measure the content of P_in_. The P_org_ content was calculated by subtracting P_in_ from P_AR_.

All soil and plant P forms as well as Fe_M3_ and Al_M3_ were measured using ICP-OES, and the organic P (P_org_) content was obtained by subtracting P_in_ from P_AR_.

#### 2.4.2. Determination of Soil P Sorption Parameters

Different soil indicators (Equations (1)–(4)) were used to analyze and evaluate P characteristics and sorption properties in chernozem soil.

The P_M3_, Al_M3_, and Fe_M3_ results were used to calculate the phosphorus sorption index (PSI_M3_), degree of P saturation (DPS_M3_ (%)), and P stability ratios on aluminum and iron (P_sta-Al_ (%) and P_sta-Fe_ (%)) according to the following equations [9,32,33]. These parameters are used to determine the risk of horizontal and vertical loss of P from agricultural land.
PSI_M3_ (mmol kg^−1^) = 0.5 [Fe_M3_+Al_M3_](1)
(2)$${\mathrm{DPS}}_{M3}\;\left(\%\right)=\frac{P_{M3}}{{\mathrm{PSI}}_{M3}}\;\times \;100$$
(3)$$P_{\mathrm{sta}-\mathrm{Al}}\;\left(\%\right)=\frac{P_{M3}}{{\mathrm{Al}}_{M3}}\;\times \;100$$
(4)$$P_{\mathrm{sta}-\mathrm{Fe}}\;\left(\%\right)=\frac{P_{M3}}{{\mathrm{Fe}}_{M3}}\;\times \;100$$
where P_sta-Al_ and P_sta-Fe_ (%) are P stability ratios on aluminum and iron, DPS_M3_ (%) is the degree of P saturation, and 0.5 is the arbitrary corrective factor used to calculate the PSI/DPS value.

### 2.5. Statistical Analysis

All the data were analyzed and evaluated using one-way analysis of variance (ANOVA) and the Pearson correlation coefficient using STATISTICA software version 13.5.0.17 (TIBCO, Paolo Alto, CA, USA). The significance of differences and mean separations were analyzed using Tukey’s HSD test at a *p* < 0.05 significance level.

## 3. Results

### 3.1. Aboveground Dry Biomass Yield (BY)

The average dry aboveground biomass yield (BY) of silage maize was evaluated from the year 1997 to 2023 after long-term fertilization with inorganic and organic fertilizers (Figure 2). The dry aboveground BY varied from 8.5 t DM ha^−1^ to 15.0 t DM ha^−1^ during 1997–2023. Significantly higher dry aboveground BY was recorded in N_240_ treatment over the years compared to other treatments (Table 3). All the treatments were significantly (*p* < 0.05) different from the control treatment (except in 2015–2017). The dry aboveground BY significantly (*p* < 0.05) decreased during 2015–2017 as well as during 2018–2020 at the fertilized treatments. This was probably due to severe drought and low precipitation in 2015 and 2018, which reduced yield across each treatment. The BY variability and reduction across the years and treatments are also associated with P uptake, which also depends on the amount of bioavailable P, N application rates, soil sickness, or fatigue due to long-term static fertilization and intensive monoculture. In addition, agronomic practices, planting densities, and the use of different maize hybrids probably contributed to yield variation because nutrient uptake particularly phosphorus and nitrogen increases with increasing nutrient availability and biomass production. Application of N_120_, N_240_, SS_120,_ and SS_240_ treatments in the growing periods (1997–2023) increased the BY by 126%, 128%, 115%, and 124%, respectively, compared to the unfertilized treatment. Total dry aboveground BY or accumulation increased in the order of Con < SS_120_ < SS_240_ < N_120_ < N_240_, treatments (Table 3). However, the overall biomass productivity showed a decreasing trend during the growing seasons.

### 3.2. P Uptake and Utilization by the Aboveground Biomass

Total and average P uptake by the aboveground biomass was not significantly different among treatments. However, the amount of P uptake increased with the nutrient application gradient. The amount of P uptake and utilization mostly depends on aboveground biomass productivity, P availability, and P utilization efficiency of the maize hybrids, particularly internal use efficiency.

The average DM and total DM yields significantly (*p* < 0.05) increased in the order of Con < SS_120_ < SS_240_ < N_120_ < N_240_ treatments across the entire year (Table 1). The N_120_ and N_240_ treatments positively influenced the DM yield compared to other treatments. This is because mineral fertilizers have temporal yield-increasing effects by providing adequate nutrient supply in the early-stage growth of a plant. However, there was DM yield variability over the years, and this is probably associated with climate variability (i.e., low precipitation and severe drought) and maize hybrids.

Similarly, the average and cumulative P balance was determined by subtracting the total P output from the total P input across the years and treatments without considering any P losses (P uptake by weeds, sorption factors, erosion, and leaching). The average yearly P uptake by plants was 20.8 kg P ha^−1^ at the control treatment. The highest P uptake was reached for both N_240_ and SS_240_ (25.1 kg P ha^−1^). A negative P balance was obtained at control, N_120,_ and N_240_ treatments (−562, −648, −678 kg P ha^−1^ 27 years^−1^, respectively), which gradually depleted the legacy P and brought a risk of P deficiency in the soil. On the other hand, a significantly positive P balance was obtained from SS_120_ and SS_240_ treatments (+1952 and +3750 kg P ha^−1^ 27 years^−1^, respectively), which eventually led to the building up of legacy P in soil.

### 3.3. Different Soil P Fractions and Sorption Parameters

During the long-term “static fertilization”, the nature of stable and bioavailable P forms and the sorption capacity of the soil were evaluated shortly after establishment (1999), in the middle (2011), and at the end of the experiment (2023). The amount of available P in P_H2O_ and P_M3_ ranged from 6.2 to 19.7 mg P kg^−1^ and 91 to 213 mg P kg^−1^, respectively (Table 4).

During 1999, there was no significant difference (*p* < 0.05) in P fractions among the treatments. The considerable difference in P_M3_, P_H2O_, P_AR_, and P_in_ was gradually visible among treatments during 2011 and 2023. All the P fractions (P_H2O_, P_M3_, P_AR_, and P_in_) were significantly (*p* < 0.05) increased during 2011 and 2023 under SS_120_ and SS_240_ treatments, whereas plots, which continuously received a high rate of N (240 kg P ha^−1^ year^−1^) fertilizer, significantly (*p* < 0.05) decreased the amount of P_M3_ compared to other treatments in 2023. This might be due to the synergistic and mobilizing effects of N fertilizers.

Similarly, the amount of P_org_ was determined between treatments and across the three abovementioned years. Although there was no significant difference among treatments in each year, low organic P content was recorded in 1999 in all treatments. However, the amount of P_org_ significantly (*p* < 0.05) increased across the years in all treatments (except SS_120_). This was probably associated with surface organic matter residue contribution and low enzyme activity in the case of Con and N fertilization while in the case of SS_120_ and SS_240_ fertilization was due to the frequent addition of organic phosphorus from the sewage sludge and breakdown and decomposition of plant residues and microbial biomass P.

After 27 years of static fertilization, the amount of pseudototal P (P_AR_) in chernozem soil increased by 4.49% (Con), 5.42% (N_120_), 1.78% (N_240_), 39.8% (SS_120_), and 46.7% (SS_240_) in each treatment at the end of the year (Table 5). Among the treatments, the highest amount of P_AR_ was recorded in SS_120_ and SS_240_ treatments representing the buildup of legacy P in the soil whereas the lowest amount of P_AR_ was recorded in the N_240_ treatment (high rate of N), which leads to the depletion of legacy P in the long run. Consequently, the amount of available P (P_M3_) in soil decreased by −6.63%, −28.9%, and −30.15% in Con, N_120_ and N_240_ and increased by 14.3% and 17.9% in SS_120_ and SS_240_ treatments, respectively.

The soil P sorption index (PSI_M3_), degree of P saturation (DPS_M3_), and P stability ratios on aluminum and iron (P_sta-Al_ and P_sta-Fe_) were calculated from the Mehlich 3 test and are presented in Table 6. All sorption parameters of topsoil (0–30 cm) were evaluated among the treatments and studied in the years 1999, 2011, and 2023, respectively.

At the beginning of the experiment (1999), there was no significant difference in soil P sorption parameters among treatments including the control. Compared to the stability ratio of the P_sta-Al_ value, a greater P_sta-Fe_ value was recorded in each treatment. The low P_sta-Al_ value obtained from this study suggests that Al was more responsible for soil P sorption as compared to the P_sta-Fe_.

In 2011, the soil sorption parameters (except PSI_M3_) showed a decreasing trend in all treatments. PSI_M3_ showed an increasing tendency in all treatments, which enhanced the sorption capacity of the soil over the years. The P_sta-Fe_ significantly (*p* < 0.05) decreased in all treatments (except N_240_). The low P_sta-Fe_ indicated that Fe is becoming more susceptible to P sorption. As a result, the amount of DPS_M3_ value in the soil decreased across all treatments.

Similarly, during 2023, all the soil sorption parameters maintained their decreasing trend across the treatments and years except the P_sta-Al_ and DPS_M3_ values in 2011 and 2023 at SS_120_ and SS_240_ treatments.

Overall, our soil had high extractable Al_M3_, contributing to high soil P sorption capacity/index (Table 6). In contrast, the DPS_M3_ value decreased across the years in each treatment except SS_240_.

The DPS_M3_ value in the studied soil ranged from 61.9 to 130% across the years. We have observed a very high DPS_M3_ value in our soil profiles, indicating a high potential for P mobility. Although there was no significant difference in DPS_M3_ among treatments, greater DPS_M3_ was recorded in SS_120_ and SS_240_ treatments across the years.

### 3.4. The Pearson’s Correlations between P Fractions and Sorption Parameters

The different P fractions determined from the post-harvested soil (Table 5) and soil sorption parameters were evaluated using Pearson’s correlation coefficient test (Table 7). The P_AR_, P_in_, P_M3_, P_H2O_, and DPS_M3_ variables were significantly correlated at the (r^2^ > 0.700; *p* < 0.01) significance level. Similarly, P_sta-Al_ was strongly correlated with (P_AR_, P_in_, P_M3_, P_H2O_, and DPS_M3_ variables) at (r^2^ > 0.700; *p* < 0.01), whereas P_sta-Fe_ was moderately correlated with DPS_M3_ at r^2^ = 0.313; *p* < 0.05. PSI_M3_ was moderately correlated with P_M3_, Fe_M3_ and P_org_ at r^2^ = 0.550, r^2^ = 0.600 and r^2^ = 0.436; *p* < 0.05 significant level, respectively. However, PSI_M3_ was strongly correlated with Al_M3_ at r^2^ = 0.948; *p* < 0.01 level.

## 4. Discussion

### 4.1. Fertilization Effects on Aboveground Dry Biomass Yield and P Uptake

This study demonstrated that the application of nitrogen (N) and sewage sludge (SS) significantly influenced maize biomass yield and P uptake. The increase in dry aboveground biomass yield (BY) followed the gradient Con < SS120 < SS240 < N120 < N240 (Table 3 and Figure 2). This trend aligns with the findings of Seghouany et al. [13], who reported substantial maize yield increments with combined N and P application compared to individual nutrient applications. According to the result in Figure 2, the amount of biomass productivity over the years showed a decreasing trend. Especially, the dry aboveground BY significantly (*p* < 0.05) decreased in all treatments including the control treatment during 2015–2017 and 2018–2020. The same BY reduction scenario was also reported in our previous studies at the Červený Újezd long-term experimental site [12,20]. One of the probable reasons was the weather conditions (extreme drought and low precipitation in the year 2015) across the entire Czech Republic [34]. This probably even contributed to agronomic practices, continuous monocultural cropping which creates soil fatigue, and the use of different maize hybrids.

The amount of P uptake, utilization, and remobilization within the plant (re-allocation, re-translocation, partitioning, and assimilation) depends on morphological and physiological conditions and P supply. For instance, about 24–42% of P is taken up at the pre-silking stage and 58–75% at the grain-filling stage because at silking 40% of the biomass is accumulated in the leave and 50% of the biomass is accumulated at the maturity stage [35]. Phosphorus is highly required by cereal crops, specifically by canola and maize plants, during silking and post-silking stages (grain-filling stages) [15]. Nutrient uptake, particularly phosphorus and nitrogen, increased with increased nutrient availability and biomass production. However, the amount of P concentration in the aboveground DM of maize increased in the order of N_120_ (1875 mg P kg^−1^) < N_240_ (1916 mg P kg^−1^) < SS_120_ (1965 mg P kg^−1^) < SS_240_ (1992 mg P kg^−1^) < Con (2039 mg P kg^−1^) treatments. The lower P content in plants on the fertilized treatments as compared to control was caused by the dilution effect. The greater P uptake exceeding plant assimilation would result in low P utilization efficiency in maize plants. The partitioning of acquired P among sink organs (roots, stems, leaves, and grains) is strongly affected by the growth and development stages of a plant (cell division and graining filling stages) and fertilizing treatments [36]. The amount of P uptake variability was associated with the aboveground dry biomass yield variability (Table 3 and Figure 2). In addition, the amount of P uptake in plants increased by N addition ≥ 150 kg N ha^−1^ [15] through increasing root branching and the length of a plant [37]. N has a dual effect on P uptake and mobilization up to 40 cm depth in soil and increased above and belowground biomass accumulation.

Total maize productivity increased across the Czech Republic from 1961 to 2010 and gradually decreased after 2010 due to high temperatures and low precipitation [34]. According to Hristov et al. [38], by 2050 climate change poses severe yield “shocks” in the EU, which even leads to crop replacement in the worst scenario compared to the 1981–2010 baseline report. The average maize biomass yield was reduced by (−9%) in 2020 and will be reduced by (−15%) in 2080 in central Europe compared to the baseline period 1971–2000 [39] because maize yield is too sensitive to high (>23 °C) and low temperatures (<10 °C) [40].

### 4.2. Soil P Evaluation

Our long-term persistent evaluation of P indicated that a high rate of N fertilization could decrease the amount of available P and P_AR_. In contrast, a high rate of organic P application (in the form of SS_120_ and SS_240_) resulted in the accumulation of excess P_M3_ and P_AR_. After 27 years of static fertilization, the amount of available P (P_M3_) in our chernozem soil significantly decreased by 6.63%, 28.9%, and 30.2% in Con, N_120,_ and N_240_ treatments, respectively, whereas the amount of pseudototal P (P_AR_) increased by 4.49%, 5.42%, and 1.78%, in Con, N_120_, and N_240_ treatments, respectively. The negative impact of N-induced soil acidification on microbial diversity and P mineralization processes further exacerbates this issue [41]. Plots that continuously received a high rate of nitrogen—N_240_ (240 kg N ha^−1^ year^−1^)—fertilizer significantly decreased the amount of bioavailable P (P_M3_) compared to other treatments that led to N-induced P deficiency in soil (Table 4). Furthermore, N application could accelerate soil acidification, which negatively affects microbial diversity and the process responsible for organic P mineralization [41,42]. Another researcher also mentioned that N application increased P uptake and mobilization by facilitating the deep rooting of a plant [7]. However, the amount of available P_M3_ fraction in our chernozem soil was greater than the recommended level including the control treatment. A similar observation has been reported by Medinsky et al. [4], where the control treatment maintains available P above the recommended level after 84 years of intensive crop rotation (sugar beet, spring barley, potato, and winter wheat) in chernozem soils. Rowe et al. [8] also mentioned that the legacy P in unfertilized soil can maintain crop yields for up to 10 years.

In contrast to N application, SS_120_ and SS_240_ treatments in our experiment showed higher available P, i.e., P_M3_-SS_120_ (181–202 mg P kg^−1^), P_M3_-SS_240_ (181–213 mg P kg^−1^), P_H2O_-SS_120_ (8.86–13.7 mg P kg^−1^), and P_H2O_-SS_240_ (8.91–19.7 mg P kg^−1^) accumulation throughout the years due to continuous mineralization and conversion of organic P. This result is consistent with Medinsky et al. [4], who reported a high amount of available P in Haplic Phaeozem soil with FYM application. The amount of pseudototal P (P_AR_) and available P (P_M3_) left as a legacy P in chernozem soil increased by 39.80% and 14.34% and by 46.72% and 17.85%, respectively, after 27 years of SS_120_ and SS_240_ application. The readily available P_H2O_ also increased in SS_120_ and SS_240_ treated soil by 54.0% and 121%, respectively. Similarly, Asrade et al. [20] reported that the amount of P_AR_ and available P_M3_ in Luvisol increased by 31% and 67%, respectively, after 27 years of sewage sludge (SS_120_) application. SS application increased available P by 64.2% [19] and by 15.9% in loamy sand soil after 18 months of application [43]. The positive phosphorus development changes in SS_120_ and SS_240_ treatments led to the buildup of legacy P in soil over time. About 64% to 73% of total P development in the soil is contributed by organic P applications [1]. Although the P in sewage sludge is highly associated with Fe and Al due to co-precipitation, long-term application of sewage sludge increased the amount of legacy P in sandy loam soil more than manure and compost [44]. Other researchers also mentioned that SS application positively influences the soil’s physical and chemical properties (increased humic acid from 1.2 to 2.2%, P content from 14 to 51mg P kg^−1^, soil pH_H2O_ from 5.5 to 5.7, cation exchange capacity from 3.2 to 4.6 cmol_+_ kg^−1^, and aluminum concentration 20–90 mg Al kg^−1^) after five years of application in podsolic soil [45].

Therefore, based on our results and other works of literature, the application of sewage sludge can completely replace mineral P fertilizer, positively influence the soil’s physical and chemical properties, and increase both available and pseudototal P (P_AR_) in all soil types.

Similarly, the amount of organic P content was quantified in all treatments during 1999, 2011, and 2023. The amount of organic P significantly (*p* < 0.05) increased across the years and treatments (except in SS_120_). Organic P is the major contributor of total P ranging from 4 to 90% in chernozem soil [46]. The P cycle (immobilization, mineralization, and transformation) processes in soil depend on the C:P ratios [47].

This study also evaluated soil P sorption parameters (Table 6), revealing that aluminum (Al) was a major controlling factor for P sorption in Chernozem soil. This finding is in line with previous studies indicating that Al significantly influences P availability and transformation in various soil types [5]. The average DPS_M3_ value in the studied soil ranged from 61.9 to 126% across the years, which is above the critical value (25%) for P leaching, the agronomic critical value of 8%, and the environmental threshold value (≥ 60%) [32]. A low P_sta-Al_ (%) value was observed in response to Con, N_120,_ and N_240_ treatments throughout the years, which indicated that Al is more responsible for the adsorption of P in the soil. This result is similar to our previous study (conducted in another experimental site), which suggested that Al was the main controlling factor for phosphorus availability and transformation in Haplic Luvisol [12,20]. This is probably associated with the transformation/degradation of chernozem soil into clay-illuviated soils (Luvisols/Retisol) since the Neolithic period in the Eastern and Central European loess belt [48].

### 4.3. Soil P Balance and Relationships

The phosphorus development change on the topsoil was evaluated after 27 years of long-term N (N_120_ and N_240_) and sewage sludge (SS_120_ and SS_240_) fertilization and maize monoculture (Table 3). A negative cumulative P balance was obtained in Con (−562 kg P ha^−1^ 27 yrs^−1^), N_120_ (−648 kg P ha^−1^ 27 yrs^−1^), and N_240_ (−678 kg P ha^−1^ 27 yrs^−1^) treatments, which approximately depleted the legacy P by −11%, −12% and −13%, respectively, compared to the initial value of P_AR_ (5247 kg P ha^−1^) in 1993 and decreased the available P_M3_ by −28.93%, −30.15%, and 6.63%, respectively. Asrade et al. [12] also reported a negative P balance in control (−483 kg P ha^−1^ 27 yrs^−1^) and in ammonium sulfate and urea ammonium nitrate (−621 kg P ha^−1^ 27 yrs^−1^) treatments in Haplic Luvisol. These led to N-induced P deficiency in soil [41]. On the other hand, a positive P balance (+1592 kg P ha^−1^27 yrs^−1^ and +3750 kg P ha^−1^ 27 yrs^−1^) was obtained from SS_120_ and SS_240_ applications at a rate of 82 kg P ha^−1^ yr^−1^ and 164 kg P ha^−1^ yr^−1^, respectively, significantly changing the P_AR_ content in soil by 30.0% and 72.0%, and available P_M3_ by 14.3% and 17.9%, respectively, compared to the initial value of P_AR_ (5247 kg P ha^−1^) in 1993. The amount of P_M3_ in each treatment across the years (Table 4) is in the range of optimal (81–115 mg P kg^−1^) and excess (>186 mg P kg^−1^) categories of the Czech Republic. In Europe, about 88% of croplands are above the threshold value of (10–25 mg Olsen P kg^−1^) for optimal crop yields [48,49,50]. The average Olsen P in our chernozem soil converted from Mehlich 3-P values using the equation (Olsen P = 12.08 + 0.25*M_3_-P) ranged from 35 to 65 mg Olsen P kg^−1^ over the years [51]. Therefore, further P fertilizer addition would increase total and available P accumulation in chernozem soil.

Pearson’s correlation coefficient analysis was used to determine the association of the variables in Table 7. A strong positive correlation was observed between P_AR_, P_in_, P_M3_, P_H2O_, and DPS_M3_ (r^2^ > 0.700; *p* < 0.001). The PSI_M3_ strongly correlated with Al_M3_ (r^2^ = 0.948; *p* < 0.001) and moderately correlated with Fe_M3_ (r^2^ = 0.600; *p* < 0.005), which implies that Al was a major controlling factor of P in soil, and this fact is supported by the findings of Kedir et al. [5].

## 5. Conclusions

Our long-term study revealed that the total aboveground maize BY increased in the order of Con < SS_120_ < SS_240_ < N_120_ < N_240_ according to the treatment application gradient. Contrastingly, after 27 years of static fertilization, the amount of available P (P_M3_) in the chernozem soil decreased by −6.63%, −28.9%, and −30.2% in Con, N_120,_ and N_240_ treatments, respectively, which gradually led to a negative P balance and N-induced P deficiency in the soil. The amount of available P_M3_ fraction was still greater than the recommended level in our chernozem soil including the control treatment, and P was not a limiting nutrient. On the other hand, the application of sewage sludge (SS_120_ and SS_240_) significantly changed the content of the pseudototal P (P_AR_) and available P (P_M3_) in our soil beyond the agronomic and crop threshold value and buildup of legacy P. According to the evaluated soil sorption parameters, Aluminum appears as the key factor influencing the P sorption and transformations. Thus, further P fertilization of the soil is not recommended. Optimal fertilization of maize for silage in the given climatic and soil conditions is achieved with the application of 240 kg N ha^−1^, ensuring high biomass yield and adequate phosphorus availability.

## Figures and Tables

**Figure 1 plants-13-02037-f001:**
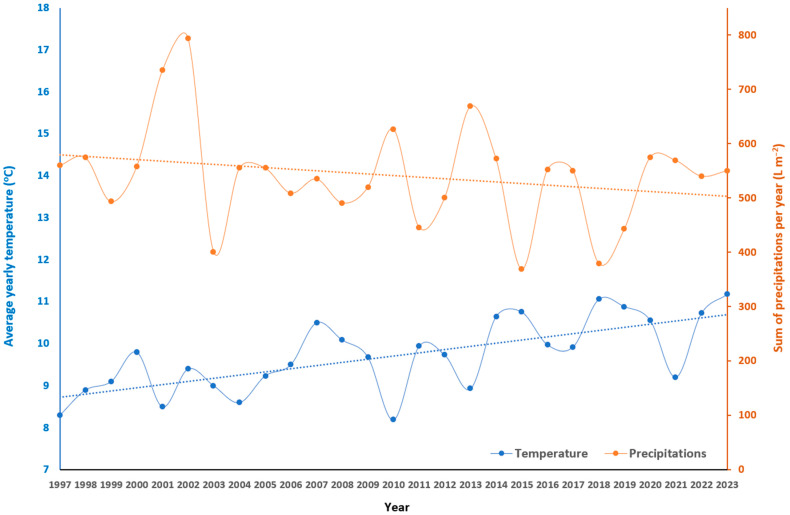
Average yearly temperatures and the sum of yearly precipitations during the experiment. Note: the data for the period 1997 to 2004 were obtained from the Czech Hydrometeorological Institute (average for Prague and Central Bohemia region) and the data from the years 2005 to 2023 were obtained from a meteorological station placed less than 100 m from the field experiment.

**Figure 2 plants-13-02037-f002:**
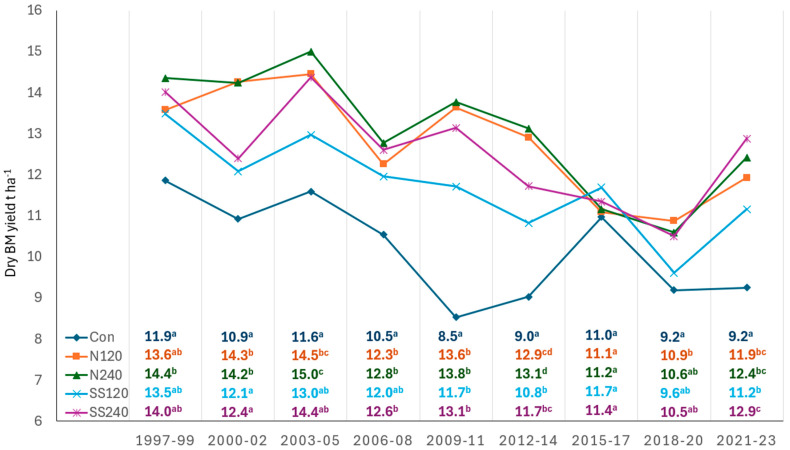
Average maize dry biomass yield (three-year periods; in t DM ha^−1^). Notes: Con, control treatment, N_120_—mineral nitrogen 120 kg N ha^−1^, N_240_—mineral nitrogen 240 kg N ha^−1^, SS_120_—sewage sludge 120 kg N ha^−1^, SS_240_—sewage sludge 240 kg N ha^−1^. Different letters are representing significant differences among treatments in the frame of the studied period (Tukey HSD test, *p* < 0.05).

**Table 1 plants-13-02037-t001:** Initial site description before the long-term field experiment commencement.

Basic Description of Soil Variables in the Site	Suchdol	References
Altitude (m a.s.l.)	286	
pH/CaCl_2_	7.5	Minasny et al. [25]
Cation exchange capacity (mmol_+_ kg^−1^)	230	
Soil type	Haplic Chernozem	NRSC USDA [26] ^1^
Soil texture	Silty loam	NRSC USDA [26] ^1^
Clay (%) (<0.002 mm)	2.2	
Silt (%) (0.002–0.05 mm)	71.8	
Sand (%) (0.05–2 mm)	26.0	
Bulk density (g cm^−3^) topsoil	1.43	
Soil depth (cm)	30	
Soil organic matter carbon C_SOM_ (%)	1.49	CNS ^2^
Total P of *Aqua regia* extraction in topsoil (0–30 cm) in mg kg^−1^ (kg ha^−1^)	1223 (5247)	ISO 11466 [27]
P_M3_ (mg P kg^−1^; year 1996)	139	Mehlich [28]

^1^ Natural Resource Conservation Service—United States Department of Agriculture. ^2^ CNS analyzer (Elementary Vario Macro, Elementar Analysensysteme, Hanau-Frankfurt am Main, Germany; after removing the carbonate C using hydrochloric acid).

**Table 2 plants-13-02037-t002:** Applied fertilizers including nitrogen and phosphorus doses.

Treatments	Avg. Dry Weight (t ha^−1^ year^−1^)	N (kg ha^−1^ year^−1^)	P (kg ha^−1^ year^−1^)	K (kg ha^−1^ year^−1^)
Con	-	-	-	-
N_120_	-	120	-	-
N_240_	-	240	-	-
SS_120_	10.17	120	82	22.4
SS_240_	20.34	240	164	44.8

Notes: Con, control; N_120_, nitrogen (120 kg ha^−1^ year^−1^); N_240_, nitrogen (240 kg ha^−1^ year^−1^); SS_120_, sewage sludge dose 120 and SS_240_, sewage sludge dose 240.

**Table 3 plants-13-02037-t003:** Total and average dry matter yields, P uptake, and P balance (1997–2023).

Treatment	DM Yield t ha^−1^ yr^−1^	Total DM Yield t ha^−1^ 27 yrs^−1^	P Uptake kg ha^−1^ yr^−1^	Total P Uptake kg ha^−1^ 27 yrs^−1^	P Balance kg ha^−1^ yr^−1^	P Balance kg ha^−1^ 27 yrs^−1^
Con	10.2 ^a^	276 ^a^	20.8 ^a^	562 ^a^	−20.8	−562
N_120_	12.8 ^c^	345 ^c^	24.0 ^a^	648 ^a^	−24.0	−648
N_240_	13.1 ^c^	352 ^c^	25.1 ^a^	678 ^a^	−25.1	−678
SS_120_	11.7 ^b^	316 ^b^	23.0 ^a^	622 ^a^	59.0	1592
SS_240_	12.6 ^bc^	339 ^bc^	25.1 ^a^	678 ^a^	139	3750

Notes: Con—control treatment, N_120_—mineral nitrogen 120 kg N ha^−1^, N_240_—mineral nitrogen 240 kg N ha^−1^, SS_120_—sewage sludge 120 kg N ha^−1^, SS_240_—sewage sludge 240 kg N ha^−1^. Different letters are representing significant differences among treatments (Tukey HSD test, *p* < 0.05).

**Table 4 plants-13-02037-t004:** The different P fractions analyzed from the post-harvest topsoil (0–30 cm).

Years	P Fractions	Treatments				
		Con	N_120_	N_240_	SS_120_	SS_240_
1999	P_M3_ (mg P kg^−1^)	139 ^aA^	155 ^aA^	135 ^aA^	181 ^aA^	181 ^aA^
	P_H2O_ (mg P kg^−1^)	6.20 ^aA^	7.70 ^aA^	6.4 ^aA^	8.9 ^aA^	8.9 ^aA^
	P_in_ (mg P kg^−1^)	1169 ^aA^	1181 ^aA^	1168 ^aA^	1327^aA^	1222 ^aA^
	P_org_ (mg P kg^−1^)	104 ^aA^	70 ^aA^	119 ^aA^	114 ^aA^	93 ^aA^
	P_AR_ (mg P kg^−1^)	1274 ^aA^	1251 ^aA^	1287 ^aA^	1289 ^aA^	1295 ^aA^
2011	P_M3_ (mg P kg^−1^)	155 ^abA^	132 ^abA^	119 ^aA^	179 ^abA^	210 ^bA^
	P_H2O_ (mg P kg^−1^)	8.7 ^aA^	7.3 ^aA^	7.2 ^aA^	12.1 ^bB^	13.0 ^bA^
	P_in_ (mg P kg^−1^)	1026 ^abA^	966 ^aA^	1031 ^abA^	1326 ^bcA^	1412 ^cAB^
	P_org_ (mg P kg^−1^)	184 ^aAB^	322 ^aB^	178 ^aA^	216 ^aA^	243 ^aB^
	P_AR_ (mg P kg^−1^)	1211 ^aA^	1288 ^abA^	1209 ^aA^	1543 ^abAB^	1656 ^bB^
2023	P_M3_ (mg P kg^−1^)	121 ^abcA^	111 ^abA^	91 ^aA^	202 ^bcA^	213 ^cA^
	P_H2O_ (mg P kg^−1^)	8.1 ^abA^	7.4 ^aA^	9.70 ^abA^	13.7 ^abB^	19.7 ^bA^
	P_in_ (mg P kg^−1^)	1023 ^aA^	967 ^aA^	967 ^aA^	1497 ^bA^	1562 ^bB^
	P_org_ (mg P kg^−1^)	289 ^aB^	356 ^aB^	341 ^aB^	305 ^aA^	338 ^aB^
	P_AR_ (mg P kg^−1^)	1312 ^aA^	1323 ^aA^	1308 ^aA^	1802 ^bB^	1900 ^bB^

Notes: P_M3_, Mehlich 3 extractable of phosphorus; P_H2O_, water extractable of phosphorus; P_in_, inorganic phosphorus; P_org_, organic phosphorus; P_AR_, *Aqua regia* extractable of pseudototal phosphorus. Con—control treatment, N_120_—mineral nitrogen 120 kg N ha^−1^, N_240_—mineral nitrogen 240 kg N ha^−1^, SS_120_—sewage sludge 120 kg N ha^−1^, SS_240_—sewage sludge 240 kg N ha^−1^. Different lowercase letters behind the values (in rows) represent significant differences among treatments (Tukey HSD test, *p* < 0.05), and different capital letters behind the values (in columns) represent significant differences among studied years (Tukey HSD test, *p* < 0.05).

**Table 5 plants-13-02037-t005:** The Pseudototal P (P_AR_) and available P (P_M3_) changes over time in the topsoil (0–30 cm).

Treatments	Pseudototal P (P_AR_) mg P kg^−1^		Changes (%)	Available P (P_M3_) mg P kg^−1^		Changes (%)
	1999	2023		1999	2023	
Con	1274 ^a^	1312 ^a^	4.49	139 ^a^	121 ^abc^	−6.63
N_120_	1251 ^a^	1323 ^a^	5.42	155 ^a^	111 ^ab^	−28.93
N_240_	1287 ^a^	1308 ^a^	1.78	135 ^a^	91.0 ^a^	−30.15
SS_120_	1289 ^a^	1802 ^b^	39.80	181 ^a^	202 ^bc^	14.34
SS_240_	1295 ^a^	1900 ^b^	46.72	181 ^a^	213 ^c^	17.85

Notes: Con—control treatment, N_120_—mineral nitrogen 120 kg N ha^−1^, N_240_—mineral nitrogen 240 kg N ha^−1^, SS_120_—sewage sludge 120 kg N ha^−1^, SS_240_—sewage sludge 240 kg N ha^−1^. Different letters are representing significant differences among treatments (Tukey HSD test, *p* < 0.05).

**Table 6 plants-13-02037-t006:** Topsoil (0–30 cm) sorption parameters and characterizations in (1999, 2011, and 2023).

Year	Soil P Sorption Parameters	Treatments				
		Con	N_120_	N_240_	SS_120_	SS_240_
1999	PSI_M3_ (mmol kg^−1^)	5.10 ^aA^	4.50 ^aA^	4.80 ^aA^	4.50 ^aA^	5.20 ^aA^
	P _sta-Fe_ (%)	457 ^aB^	520 ^aB^	464 ^aA^	585 ^aB^	493 ^aB^
	P _sta-Al_ (%)	54.1 ^aA^	60.4 ^aA^	55.8 ^aA^	79.7 ^aA^	70.0 ^aA^
	DPS_M3_ (%)	89.1 ^aA^	99.6 ^aA^	91.8 ^aA^	130 ^aA^	114 ^aA^
2011	PSI_M3_ (mmol kg^−1^)	6.10 ^aA^	5.80 ^aA^	5.20 ^aA^	5.20 ^aA^	5.90 ^aA^
	P _sta-Fe_ (%)	313 ^aAB^	311 ^aA^	328 ^aA^	311 ^aA^	277 ^aA^
	P _sta-Al_ (%)	52.0 ^abA^	46.6 ^abA^	45.7 ^aA^	71.0 ^abA^	76.0 ^bA^
	DPS_M3_ (%)	83.7 ^aA^	75.6 ^aA^	74.6 ^aA^	111 ^aA^	117 ^aA^
2023	PSI_M3_ (mmol kg^−1^)	5.70 ^aA^	5.40 ^aA^	4.80 ^aA^	5.20 ^aA^	5.60 ^aA^
	P _sta-Fe_ (%)	262 ^aA^	274 ^aA^	302 ^aA^	281 ^aA^	255 ^aA^
	P _sta-Al_ (%)	43.5 ^aA^	41.5 ^aA^	37.7 ^aA^	82.4 ^bA^	80.5 ^bA^
	DPS_M3_ (%)	70.1 ^aA^	67.3 ^aA^	61.9 ^aA^	126 ^bA^	121 ^bA^

Notes: PSI_M3,_ soil phosphorus sorption index; DPS_M3_ (%), degree of phosphorus saturation; P_sta-Al_ (%), and P_sta-Fe_ (%), phosphorus stability ratios on aluminum and iron, respectively; Con—control treatment, N_120_—mineral nitrogen 120 kg N ha^−1^, N_240_—mineral nitrogen 240 kg N ha^−1^, SS_120_—sewage sludge 120 kg N ha^−1^, SS_240_—sewage sludge 240 kg N ha^−1^. Different lowercase letters behind the values (in rows) represent significant differences among treatments (Tukey HSD test, *p* < 0.05), and different capital letters behind the values (in columns) represent significant differences among studied years (Tukey HSD test, *p* < 0.05).

**Table 7 plants-13-02037-t007:** The Pearson’s correlation coefficients among P fractions and sorption parameters.

	P_M3_	Fe_M3_	Al_M3_	P_H2O_	P_in_	P_org_	P_AR_	PSI_M3_	DPS_M3_	P_sta-Fe_	P_sta-Al_
P_M3_ (mg kg^−1^)	1.00										
Fe_M3_ (mg kg^−1^)	0.917 **	1.00									
Al_M3_ (mg kg^−1^)	0.288	0.314 *	1.00								
P_H2O_ (mg kg^−1^)	0.465 *	0.495 *	−0.387 *	1.00							
P_in_ (mg kg^−1^)	0.807 **	0.782 **	−0.271	0.801 **	1.00						
P_org_ (mg kg^−1^)	0.202	0.160	0.454 *	−0.162	−0.115	1.00					
P_AR_ (mg kg^−1^)	0.851 **	0.817 **	−0.176	0.768 **	0.978 **	0.096	1.00				
PSI_M3_ (mmol g^−1^)	0.550 *	0.600 *	0.948 **	−0.160	0.034	0.436 *	0.126	1.00			
DPS_M3_ (%)	0.860 **	0.747 **	−0.197	0.672 *	0.956 **	0.048	0.948 **	0.085	1.00		
P_sta-Fe_ (%)	0.178	−0.204	−0.163	0.026	0.123	0.145	0.154	−0.206	0.313 *	1.00	
P_sta-Al_ (%)	0.859 **	0.791 **	−0.196	0.713 **	0.977 **	−0.060	0.966 **	0.101	0.993 **	0.209	1.00

Note P_M3_, Mehlich 3 extractable of P; Fe_M3_, Mehlich 3 extractable of Fe; Al_M3_, Mehlich 3 extractable of Al; P_H2O_, water extractable of P; P_in_, inorganic P; P_org_, organic P; P_AR_, *Aqua regia* extractable of pseudototal P; PSI_M3,_ soil P sorption index; DPS_M3,_ degree of P saturation; P_sta-Al_ and P_sta-Fe_, P stability ratios on aluminum and iron, respectively; * correlation is significant at *p* < 0.05; ** correlation is significant at *p* < 0.01 significant level (*n* = 21).

## Data Availability

The data that support the findings of this study are available from the corresponding author upon reasonable request.

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
