# Peer review of "Phosphorus Availability and Balance with Long-Term Sewage Sludge and Nitrogen Fertilization in Chernozem Soil under Maize Monoculture"

_plants, 2024, doi:10.3390/plants13152037_

Round 1

Reviewer 1 Report

Comments and Suggestions for Authors

Comments and Suggestions for Authors

Title: Phosphorus availability and balance with long-term sewage sludge and
nitrogen fertilization in Chernozem soil under maize monoculture

Dear Authors

The scope of research results presented in the manuscript falls within the publishing profile of Plants journal. The aim of this study was to assess the P development change and N interactive effects on phosphorus mobility and transformation in Chernozem soil after long-term mineral and sewage sludge fertilization of silage maize monoculture. The research results are interesting. The manuscript is well written. The obtained results have been well described. In order to increase the usefulness of the article, Authors must refer to the following points.

Remarks:

v  Results: A table should be added with data on the dry matter content in the above-ground parts of maize grown for silage. Lines 195-196 You need to add the depth (0-20 cm or 0-30 cm). Table 5 needs to be technically improved.

v  Discussion: Line 251 According to the data presented in the M&M section, the long-term experience lasted shorter.

v  Materials and Methods: Meteorological data should be added for the years of research. The data presented in Table 6 should be described. Was potassium fertilization used? If so, please provide the dose of potassium and the form of fertilizer. Line 404 - Is sewage sludge organic fertilizer or waste? The dose of sewage sludge, its chemical composition and its source should be provided. Conclusions: It is necessary to propose optimal fertilization of maize for silage in given climatic and soil conditions.

Specific remarks:

v  Reference 44 is not cited in the MS text. The technical quality of the mathematical formulas in subsection 4.4.2 can be improved.

Best regards

Author Response

Dear reviewer 1

We sincerely thank for your insightful comments and constructive feedback. Your expertise and suggestions have greatly enhanced the quality and clarity of our manuscript. We have carefully addressed all the comments and made the necessary revisions to improve the content. We appreciate the time and effort you dedicated to evaluating our work. Below you can see point by point reactions on your valuable comments:

Comment 1: A table should be added with data on the dry matter content in the above-ground parts of maize grown for silage.

Reaction 1: Thanks for thos comment - for better transparency, we added the values of maize dry biomass yield in the figure 1. (page 7 of clean version o resubmitted manuscript)

Comment 2: Lines 195-196 You need to add the depth (0-20 cm or 0-30 cm). 

Reaction 2: We added the soil depth in the figure caption. We did the changes accordingly even in other related tables.

Comment 3: Table 5 needs to be technically improved.

Reaction 3: We improved the technical quality of table 5. according to the journal formatting requests. Sorry for previous older version of table. (corrected table is on page 12 of clean version o resubmitted manuscript)

Comment 4: Discussion: Line 251 According to the data presented in the M&M section, the long-term experience lasted shorter.

Reaction 4: Sorry for the misunderstanding. The 55 years of the experiment was meaning the duration of related research of other authors discussed later. We did the changes to avoid this misunderstanding.

Comment 5: Meteorological data should be added for the years of research.

Reaction 5: We prepared the figure with the meteorological characteristics (page 4 of clean revised version) including linear regression showing the increasing temperature during the experiment as well as decreasing precipitations.

Comment 6: The data presented in Table 6 should be described

Reaction 6: We described the data in table 6. (Table 1. in corrected version) as well as the tendencies in newly created figure 1.

Comment 7: Was potassium fertilization used? If so, please provide the dose of potassium and the form of fertilizer. 

Reaction 7: In both, N120 and N240 treatments, potassium was not applied, because of relatively high content of K in soil. Using sewage sludge, 22.4 and 44.8 kg of K per year was applied, respectively. We added this information to the table 2. (Page 5 of clean revised version).

Comment 8: Is sewage sludge organic fertilizer or waste?

Reaction 8: Sorry for the wrong expression. Sewage sludge is according to the Czech legislative considered as waste material. We did the changes accordingly and used instead of organic fertilizer the term "sewage sludge" only.

Comment 9: The dose of sewage sludge, its chemical composition and its source should be provided.

Reaction 9: The dose of sewage sludge dry mass is mentioned in table 2. (Page 5). We added even information about aluminum and iron content applied with sludge, because these elements are responsible for phosphorus sorption. However, according to the confidentiality agreement with waste water treatment plant (WWTP), we cannot provide the information about the source. We can give only basic information that WWTP is processing the water from more than 100000 people agglomeration and the sludge is processed anaerobically.

Comment 10: It is necessary to propose optimal fertilization of maize for silage in given climatic and soil conditions.

Reaction 10: We added following recommendation at the end of conclusion chapter: It is necessary to propose optimal fertilization of maize for silage in given climatic and soil conditions.

Comment 11: Reference 44 is not cited in the MS text. 

Reaction 11: Sorry for this mistake. We changed the references almost completely even after the 2nd reviewer comments.

Comment 12: formulas in subsection 4.4.2 can be improved.

Reaction 12: The formatting of formulas in subsection 4.4.2 (2.4.2. in corrected version) was improved.

Reviewer 2 Report

Comments and Suggestions for Authors

The paper is written relatively correctly. The objectives for the implementation of the experiment are correct. Although the issues discussed have been studied for a very long time. The paper is marked with 15 remarks, easy to correct. The discussion should be supplemented with elements comparing with the results of other studies. To say the conclusion of the studies and give a number from references in brackets is not enough.

Author Response

Dear reviewer 2,

We extend our heartfelt gratitude your valuable comments and thorough review of our manuscript. Your detailed feedback has been instrumental in refining our study. We have incorporated your suggestions to strengthen the overall quality and presentation. Thank you for your thoughtful contributions and commitment to improving our research. You can find point by point reaction on your comments below:

Comment 1: The discussion should be supplemented with elements comparing with the results of other studies. To say the conclusion of the studies and give a number from references in brackets is not enough.

Reaction 1: Thanks for this suggestion. We tried to improve the discussion and its text flow. we hope we fulfilled your requirements. We added even the recommendation for fertilizing to the conclusion.

Comment 2: According to the magazine's editorial policy. The abstract should not exceed 200 words. Instructions to authors should be read.

Reaction 2: Sorry for this mistake. We shortened the abstract on 188 words.

Comments 3 and 6: mistakes in spaces.

Reaction: We went through manuscript few more times and did the correction of typing errors,

Comment 4: Introduction: Lines 50-55. Is this information needed here? This is not directly related to the topic of the study.

Reaction 4: This information was really not directly related to our research. We deleted it. Thanks for the suggestion.

Comment: 5: Here should be a chapter on methodology (in front of results, discussion and conclusions). Please move it.

Reaction 5: We followed the journal rules, where methodology should be after conclusions. However, we strongly agree with you that according to the logical structure, methodology is better in front of results. We therefore moved methodology in front of results chapter.

Comment 7: missing description of axis in the figure 1:

Reaction 7: We added the description as well as the values of maize aboveground dry biomass yield as requested from reviewer 1

Comment 8: Where such years come from. (Wrong interpretation of experiment duration.)

Reaction 8: Sorry for this misunderstanding. The 55 years of the experiment was meaning the duration of related research of other authors discussed later. We did the changes to avoid this misunderstanding.

Comment 9: Missing reference of NRSC USDA

Reaction 9:  References were added

Comment 10: Give the meaning of the abbreviations SS.

Reaction 10: The meaning of the abbreviations was added

Comment 11: Please indicate the period of the experiment.

Reaction 11: The period (27 years) of evaluated part of experiment was added in the text (line 116 in clean version of corrected manuscript)

Comment 12: Why such strange literature items on methodology. In the European Union there are corresponding standards.

Reaction 12: We added more relevant reference for dry biomass analysis, which we used for P determination (Mader, P.; Száková, J.; Miholová, D. Classical dry ashing of biological and agricultural materials. Part II: Losses of analytes due to 584 their retention in an insoluble residue. Analusis, 1998, 26, 121-129). Thanks for this suggestion.

Comment 13: An old-timer from half a century ago. There are more recent studies.

Reaction 13: We added the reference on more recent study (year 2009), where we used the same extraction procedure.

Comment 14: Explain these abbreviations. (FeM3 and AlM3)

Reaction 14: Abbreviations were explained (lines 159-160 in corrected clean version)

Comment 15: Correct the notation. [9] [39,52]

Reaction 15: Notation was corrected

Comment 16: Titles of journals should be given as abbreviations.

Reaction 16: We corrected the journal names on the abbreviated versions.

Thanks again for your valuable review

Yours sincerely

Authors
